# Cytisine-Pterocarpan Derived Compounds: Biomimetic Synthesis and Apoptosis-Inducing Activity in Human Breast Cancer Cells

**DOI:** 10.3390/molecules23123059

**Published:** 2018-11-22

**Authors:** Ting-Ting Peng, Xuan-Rong Sun, Ren-Hao Liu, Lu-Xia Hua, Dong-Ping Cheng, Bin Mao, Xing-Nuo Li

**Affiliations:** 1College of Pharmaceutical Science, Zhejiang University of Technology, Hangzhou 310014, China; pengtingting1993@163.com (T.-T.P.); liurenhao1994@outlook.com (R.-H.L.); hlhua94@163.com (L.-X.H.); chengdp@zjut.edu.cn (D.-P.C.); 2Collaborative Innovation Center of Yangtze River Delta Region Green Pharmaceuticals, Zhejiang University of Technology, Hangzhou 310014, China; sunxr@zjut.edu.cn (X.-R.S.); maob@zjut.edu.cn (B.M.)

**Keywords:** cytisine-pterocarpan derived compounds, biomimetic synthesis, breast cancer, mitochondrion-mediated apoptosis

## Abstract

Cytisine-pterocarpan derived compounds were biomimetically synthesized with (-)-cytisine and (-)-maackiain via a *N*,*N*-4-dimethyl-4-aminopyridine (DMAP)-mediated synthetic strategy in a mild manner. In the present study, tonkinensine B (**4**) was elaborated in good and high yields with the optimized reaction conditions. The in vitro cytotoxicity of compound **4** was evaluated against breast cancer cell lines and showed that **4** had a better cytotoxicity against MDA-MB-231 cells (IC_50_ = 19.2 μM). Depending on the research on cytotoxicities of **4** against RAW 264.7 and BV2 cells, it was suggested that **4** produced low cytotoxic effects on the central nervous system. Further study indicated that **4** demonstrated cytotoxic activity against MDA-MB-231 cells and the cytotoxic activity was induced by apoptosis. The results implied that the apoptosis might be induced by mitochondrion-mediated apoptosis via regulating the ratio of Bax/Bcl-2 and promoting the release of cytochrome c from the mitochondrion to the cytoplasm in MDA-MB-231 cells.

## 1. Introduction

The most commonly diagnosed cancer among women in America is breast cancer [1]. It has become the second leading cause of cancer-related deaths among women in the USA. Although treatment for cancers have been developed and much progress has been made to understand its biology, cancer is still a serious threat to human health worldwide. In China, according to the most updated but limited cancer registries, breast cancer is the fifth leading cause of cancer-related death for females [2]. Many patients with triple-negative breast cancer (TNBC), which lacks the estrogen receptor (ER), progesterone receptor (PR), and human epidermal growth factor receptor-2 (HER2), have a high death rate in Africa [3]. These three types of receptors are major valuable targets of chemotherapy [4].

Even though there are many therapeutic strategies for human breast cancer, many patients lack a durable response to these treatments [5]. Designing and developing new bioactive molecules is one of the most important research areas in the field of medicinal chemistry. Therefore, there is an urgent need to investigate the molecular mechanisms and to explore novel diagnostic and therapeutic strategies for TNBC [6].

Apoptosis is a self-regulatory process, mediated by caspases and some other self-controlled mechanisms. Apoptosis is usually activated by chemotherapeutic drugs and it is one of the major mechanisms to eliminate cancer cells without eliciting damage to normal cells or surrounding tissues [7]. In particular, the mitochondrion-mediated intrinsic pathway hold the dominate statue in mammalian cells apoptosis [8]. Mitochondrion fulfill important functions in programmed cell death [9]. While the extrinsic pathway is triggered by ligand-receptor interaction, the intrinsic pathway is initiated by intracellular processes, such as cell stress and leads to the release of cytochrome c from the mitochondria. The cytochrome in turn activates apoptosis through a number of cascade reactions in the cell [10]. A study [11] showed that deoxybouvardin can induce mitochondria-mediated apoptosis and exert strong anti-tumor activity against human breast cancer. Thus, targeting apoptosis pathways in premalignant and malignant cells is an effective strategy for cancer prevention and treatment.

Tonkinensine A (**3**) and tonkinensine B (**4**) have attracted our great interest because their structures were consisted of (-)-cytisine (**1**) and (-)-maackiain (**2**), **1** is a naturally occurring lupin alkaloid. The pharmacological property of **1**, like nicotine, is the typical ganglionic toxin that stimulates the central nervous system and autonomic nervous system [12]. It also can stimulate the local excitation of neuronal nicotinic acetylcholine receptors [13], and the certain hypolipidemic capacity of **1** has been confirmed [14].

(-)-Maackiain (**2**) is an isoflavone with high antimycotic and antibacterial properties. It usually suppresses transcriptional upregulation of the histamine H1 receptor and *interleukin-4* genes in Kujin [15]. In addition, **2** can also inhibit the cytopathic effect of HIV-1 and HIV-1 reverse transcriptase [16].

Tonkinensine **A** and **B** were the first report of the existence of cytisine-type alkaloids that feature the skeleton with a linkage to the pterocarpan and were tested for cytotoxicity against human breast cancer cells [17]. They were obtained from the extract of *Sophora tonkinensis* which is a traditional Chinese medicine [18]. Growing evidence has shown that many herbal medicines possess anticancer activity and are considered as important alternatives to the conventional treatments for cancers [19]. As the continuous interest in the development of active potential agents in breast cancer cells, the novel cytisine-pterocarpan derived compounds (Figure 1) might be a promising inhibitor for the treatment of breast cancer. Unfortunately, the low content of **3** and **4** in the extract of *S.tonkinensis* limited its clinical application. Therefore, we focused on the biomimetic synthesis of **3** and **4** as novel anticancer agents.

## 2. Results and Discussion

### 2.1. Synthsis of Compound ***4*** with ***1*** and ***2***

In this paper, we adopted the biomimetic synthesis strategy as guidance of later bio-activities assays. (-)-Cytisine (**1**)was linked with **2** by methylation and oxidative coupling reactions. According to the preliminary bio-activity assays, and compound **4** was found to inhibit the proliferation of cancer cells, especially for TNBC. In order to obtain a high content of target compounds in a mild and economic manner for further evaluation in anticancer assays, we poured our attention to the C–N coupling strategy about inexpensive catalyst of *N*,*N*-4-dimethyl-4-aminopyridine (DMAP) [20,21,22].

Both **1** and **2** were chosen as reaction materials, and the reaction was performed in the presence of DMAP and formaldehyde (HCHO) (see Scheme 1) [23,24,25]. Delightfully, the desired products **3** and **4** were obtained with 60% total yield when 2-propanol was used as a reaction solvent. The synthetic route provided further evidence for the previous conjectures of biomimetic synthetic pathway.

Optimization of the reaction conditions was performed for the synthesis of the target products. Such as the reaction solvent and the ratio of **1** and **2** were also tested, The reaction of **1** and **2** in the presence of 1,4-dioxane as solvent afforded target products in a high yield of 74% (see Table 1). Structures of products **3** and **4** were confirmed by analysis of their spectral data (^1^H, ^13^C NMR and Electrospray Ionization–MS).

### 2.2. Cytotoxic Activity In Vitro against Select Cell Lines

The reaction was firstly conducted to afford a large number of target products with the optimized reaction conditions. And cytotoxicities of **1**, **2**, and the newly prepared compound **4** were tested against different cell lines by the 3-(4,5-Dimethylthiazol-2-yl)-2,5-diphenyltetrazolium bromide (MTT) assay in vitro (Figure 2). The tumor cell lines consisted of the MCF-7 cell and the MDA-MB-231 cell. The results of bioactivities in terms of IC_50_ values are shown in Table 2. In addition, the IC_50_ values of the compounds with inhibitory activities higher than 50% were further determined. In addition, doxorubicin was chosen as the control agent. The newly prepared compound **4** (IC_50_ = 19.2 μM) had a better anti-proliferation effect than **1** (IC_50_ = 58.2 μM) and **2** (IC_50_ = 31.2 μM) on the MDA-MB-231 cell line, and had a better anti-proliferation effect than the MCF-7 cell line (IC_50_ = 31.4 μM). We further evaluated the cytotoxicity of **4** on the RAW 264.7 and BV2 cell lines, Raw 264.7 is a murine macrophage-like cell line commonly used to study the antiinflammatory effect of natural compounds [26]. As the principal resident macrophages in the central nervous system, BV2 is the first line of defense against pathological infection and play an important role in regulating neuroinflammation [27]. Importantly, **4** had lower toxicity to RAW 264.7 and BV2 cells than the standard drugs, doxorubicin. The distinct cytotoxicities of **4** between cancer cells and normal cells attracted our interest to make further biological studies.

In vitro cytotoxicity of **1**, **2** and **4** against two cancer cell lines and two non-cancerous cell lines. The data represent the mean ± SD of three individual experiments.

### 2.3. The Effect of Compound **4** on Apoptosis Induction in MDA-MB-231 Cell

Apoptosis which is a type of cell death is widely recognized as being of great importance in cancer treatment [28]. To investigate the underlying mechanisms which were responsible for the distinct cytotoxicity, further studies on the effects of **4** on cellular apoptosis were carried out.

Annexin V-Fluorescein Isothiocyanate (FITC) / propidium iodide (PI) double staining and flow cytometry were used to quantitatively analyze the ability of **4** to induce apoptosis in MDA-MB-231 cells [29]. As a result, after the treatment of **4** (6.25, 12.5 μM) for 48 h, **4** significantly increased late apoptosis of MDA-MB-231 cells. The apoptosis rates of MDA-MB-231 cells were 5.76 ± 2.12% and 14.88 ± 3.37%, respectively, which were significantly higher than that of the control group (3.86 ± 1.42%, ***p* < 0.01) (Figure 3).

### 2.4. The Effect of Compound 4 on Mitochondrion-Mediated Apoptosis in MDA-MB-231 Cell

The effect of **4** on apoptotic proteins in the mitochondrion-mediated apoptotic signaling pathway were investigated by Western blot analysis. After treatment with **4** (6.25, 12.5 μM) for 12 h, total proteins, cytoplasmic, and mitochondrial proteins were extracted to determine the content of these proteins in MDA-MB-231 cells.

The total protein was used to investigate the content of Bcl-2 and Bax protein levels in MDA-MB-231 cell. As shown in Figure 4, The expression of Bcl-2 protein was reduced and the expression of Bax protein increased in MDA-MB-231 cells with the treatment of compound **4** compared to the control. Meanwhile, the result of the ratio of Bax/Bcl-2 increased significantly compared to the control in MDA-MB-231 cells.

On the basis of the above-described experimental results, compared to the control group, the content of cytochrome c in mitochondrion was reduced, meanwhile, the content of cytochrome c in cytoplasm was significantly increased in MDA-MB-231 cells. The results illustrated that cytochrome c was released from mitochondrion to the cytoplasm with the treatment of **4** in MDA-MB-231 cell.

## 3. Materials and Methods

### 3.1. Materials

Reagents and solvents used in the synthesis of cytisine-pterocarpan derived compounds were procured commercially and used without further purification, unless otherwise indicated. Progress of the reaction was monitored by thin-layer chromatography (TLC) on pre-coated silica gel GF254 plates (Qingdao Haiyang Chemical Co. Ltd., Qingdao, China), and the spots were visualized under UV light. Silica gel (200–300 mesh) (Qingdao Haiyang Chemical Co. Ltd., Qingdao, China) was used for column chromatography, further purified on semi-preparative HPLC (Lumtech K501 series with a Lumtech K2501 UV spectrophotometer) to obtain products. The NMR (^1^H and ^13^C) spectra were recorded on a Bruker Avance at 400 MHz, using deuterated solvents (CDCl3) and tetramethylsilane as an internal standard. Chemical shifts are reported in parts per million (ppm) and J values are reported in hertz (Hz). Multiplicities are reported as follows: singlet (s), doublet (d), triplet (t), multiplet (m). The ESI-MS was measured on a Finnigan LCQ DECA instrument.

### 3.2. Synthesis of Compound **4**


To a solution of **1** (0.4 mmol) and **2** (0.2 mmol) in 1,4-dioxane (2 mL), DMAP (0.004 mmol, 0.5 mg) and HCHO (0.4 mmol, 37–40%) were added. The reaction solvent was stirred and circumfluence, the completion of the reaction was monitored by TLC (dichloromethane/methyl alcohol = 30:1). The organic solvent was removed under reduced pressure and the crude products **3** and **4** were obtained by column chromatography on silica gel (dichloromethane/methyl alcohol = 30:1). further purified on semi-preparative HPLC (Venusil XBP C18 reversed-phase column, 10 μm, 250 × 10 mm, CH_3_OH/H_2_O = 70:30) to obtain the pure product of **4**. The purity of **4** (>99%) was quantified by HPLC analysis with a Waters C-18 column (250 × 4.6 mm) and photodiode array (PDA) detector using a flow rate of 1 mL/min. and a gradient of 0.01% TFA in acetonitrile.

Tonkinensine A (**3**): (13% yield) ^1^H-NMR (CDCl_3_, 400 MHz) δ: 7.29 (dd, *J*_1_ = 6.8 Hz, *J*_2_ = 9.2 Hz, H-4), 7.04 (s, H-1′), 6.70 (s, H-7′), 6.51 (dd, *J*_1_ = 1.2 Hz, *J*_2_ = 9.2 Hz, H-3), 6.41 (s, H-10′), 6.31 (s, H-4′), 5.97 (dd, *J*_1_ = 1.2 Hz, *J*_2_ = 6.8, H-5), 5.91 (d, *J* = 1.6 Hz, -OCH_2_O-), 5.88 (d, *J* = 1.6 Hz, -OCH_2_O-), 5.38 (d, *J* = 6.4 Hz, H-11′α), 4.16 (dd, *J*_1_ = 4.8 Hz, *J*_2_ = 11.2 Hz, H-6′), 4.11 (br d, *J*=15.6 Hz, H-10α), 3.88 (dd, *J*_1_ = 6.4 Hz, *J*_2_ = 15.6Hz, H-10β), 3.61 (d, *J* = 13.6 Hz, H-14β), 3.51 (d, *J* = 13.6 Hz, H-14α), 3.55 (t, *J* = 11.2 Hz, H-6′), 3.39 (m, H-6′α), 3.11 (br d, *J* = 11.6 Hz, H-11β), 3.05 (m, H-7), 3.03 (m, H-13β), 2.51 (br s, H-9), 2.45 (br d *J* = 8.8 Hz, H-13α), 2.34 (br d, *J* = 11.6 Hz, H-11α), 1.97 (br d, *J* = 12.8 Hz, H-8β), 1.86 (br d, *J* = 12.8 Hz, H-8α); ^13^C-NMR (CDCl_3_, 100 MHz) δ: 163.4 (s), 159.0 (s), 156.4 (s), 154.1(s), 149.4 (s), 148.1 (s), 141.7 (s), 138.7 (d), 130.6 (d), 118.1 (s), 117.7 (d), 115.2 (s), 110.7 (s), 104.9 (d), 104.8 (d), 104.4 (d), 101.3(t), 93.7 (d), 78.7(d), 66.3 (t), 60.9 (t), 60.7 (t), 59.4 (t), 49.5 (t), 40.2 (d), 35.1(d), 27.6 (d), 25.8 (t).

Tonkinensine B (**4**): (61% yield) ^1^H-NMR (CDCl_3_, 400 MHz) δ: 7.28 (dd, *J*_1_ = 6.8 Hz, *J*_2_ = 9.2 Hz, H-4), 7.21 (d, *J* = 8.4 Hz, H-1′), 6.69 (s, H-7′), 6.52 (dd, *J*_1_ = 1.2 Hz, *J*_2_ = 9.2 Hz, H-3), 6.44 (d, *J* = 8.4 Hz, H-2′), 6.40 (s, H-10′), 5.96 (dd, *J*_1_ = 1.2 Hz, *J*_2_ = 6.8, H-5), 5.90 (d, *J* = 1.6 Hz, -OCH_2_O-), 5.88 (d, *J* = 1.6 Hz, -OCH_2_O-), 5.42 (d, *J* = 6.4 Hz, H-11′α), 4.17 (dd, *J*_1_ = 4.8 Hz, *J*_2_ = 11.2 Hz, H-6′), 4.13 (br d, *J* = 16.0 Hz, H-10α), 3.89 (dd, *J*_1_ = 6.4 Hz, *J*_2_ = 16.0 Hz, H-10β), 3.70 (d, *J* = 14.4 Hz, H-14β), 3.66 (d, *J* = 14.4 Hz, H-14α), 3.55 (t, *J* = 11.2 Hz, H-6′), 3.40 (m, H-6′α), 3.10 (br d, *J* = 11.2 Hz, H-11β), 3.02 (m, H-7), 2.99 (m, H-13β), 2.50 (br s, H-9), 2.44 (m, H-13α), 2.39 (br d, *J* = 11.2 Hz, H-11α), 1.95 (br d, *J* = 12.8 Hz, H-8β), 1.84 (br d, *J* = 12.8 Hz, H-8α) ^13^C-NMR (CDCl_3_, 100 MHz) δ: 163.5 (s), 159.4 (s), 154.3 (s), 153.8 (s), 149.3 (s), 148.1 (s), 141.6 (s), 138.7 (d), 130.7 (d), 118.0 (s), 117.6 (d), 110.7 (d), 110.6 (s), 108.0 (s), 105.1 (d), 104.6 (d), 101.2 (t), 93.8 (d), 79.0 (d), 66.6 (t), 60.5 (t), 59.5 (t), 53.6 (t), 49.5 (t), 40.0 (d), 35.0 (d), 27.7 (d), 25.8 (t).

### 3.3. Biological Studies

#### 3.3.1. Cell Lines, Cell Culture

Mouse macrophage cell lines (RAW 264.7), mouse microglia lines (BV2), and human breast cancer cell lines (MCF-7) were purchased from the Shanghai Institute of Cell Biology (Shanghai, China). Cells were maintained in Roswell Park Memorial Institute medium 1640 (HyClone, Logan, UT, USA), supplemented with 10% fetal bovine serum (GIBCO, Grand Island, Empire State, USA) and penicillin/streptomycin (100 U/mL and 100 mg/mL, resp.) at 37 °C under a humidified atmosphere of 5% CO_2_. Human breast carcinoma cell line (MDA-MB-231) was purchased from the Shanghai Institute of Cell Biology (Shanghai, China). Cells were maintained in Leibovitz’s L-15 (GIBCO, Grand Island, NY, USA), supplemented with 10% fetal bovine serum (GIBCO, Grand Island, NY, USA) and penicillin/streptomycin (100 U/mL and 100 mg/mL, respectively) at 37 °C in a closed incubator without CO_2_.

#### 3.3.2. Cytotoxicity Assay In Vitro

Test solutions of the compounds tested (20 mM) were prepared by dissolving the substances in DMSO (Sigma–Aldrich, Munich, Freistaat Bayern, Germany). The in vitro cytotoxic effect of all agents was examined using the MTT (MV4-11) assay. The results were calculated as IC_50_ (inhibitory concentration 50%) of the concentration of tested agent, which is cytotoxic for 50% of cancer cells. The IC_50_ values were calculated for each experiment separately, and mean values ± SD are presented in Table 2. Each compound in each concentration was tested in triplicate in a single experiment, which was repeated 3–5 times.

#### 3.3.3. MTT Assay

RAW 264.7, BV2, MCF-7, and MDA-MB-231 cell lines were seeded in 96-well plates at a density of 1 × 10^4^ cells per well. After cultured overnight, cells were treated with **1**, **2**, and **4** with different concentrations for 48 h, and 20 μL of MTT (5 mg/mL, in PBS) was added to each well and incubated for another 4 h. The MTT containing media was aspirated and the formazan crystals were dissolved using 150 μL of DMSO. The absorbance in 570 nm was detected by a microplate reader (BioTek synergy HT, Winooski, VT, USA). Cells treated with medium with 0.2% DMSO was regarded as control groups, IC_50_ value was determined by software GraphPad prism.

#### 3.3.4. Annexin V-FITC/ Propidium Iodide Analysis

The MDA-MB-231 cells were seeded in 6-well culture plates and were cultured in the absence or presence of **4** for 48 h. The cells were washed twice with cold phosphate buffer saline (PBS) and then resuspend in 1 × Binding Buffer at a concentration of 1 × 10^6^ cells/mL. Then, the cells were transferred in 100 μL of the solution (1 × 10^5^ cells) to a 5 mL culture tube. Five μL of FITC Annexin V and 5 μL propidium iodide was added. The cells were gently vortexed and incubated for 15 min at room temperature (25 °C) in the dark. Four-hundred μL of 1× Binding Buffer was then added to each tube. The flow cytometry was then analyzed within 1 hr. Analysis was performed by flow cytometry and analyzed using Lysis software (EPICS-XL, Ramsey, MN, USA). 

#### 3.3.5. Western Blot Analysis

Total proteins were prepared with radio-immunoprecipitation assay (RIPA) lysis buffer (Beyotime biotechnology, Shanghai, China). Cytoplasmic and mitochondrial proteins were extracted from MDA-MB-231 cells using Cytoplasmic and Mitochondria Protein Extraction Kit (Sangon Bio-Tech, Shanghai, China) to determine the levels of cytochrome c in both the cytoplasm and mitochondrion of MDA-MB-231 cells. Protein concentrations were measured by a bicinchoninic acid (BCA) Protein Assay Kit (Beyotime biotechnology, Shanghai, China) using bovine serum albumin (BSA) as the standard. A total of 50 mg of protein samples from each cell line was subjected to 12% sodiumdodecyl sulfate-polyacrylamide gel electrophoresis (SDS-PAGE) and transferred onto polyvinylidene difluoride (PVFD) membrane. After regular blocking and washing, the membranes were incubated with primary antibodies overnight at 4 °C, followed by incubating with horizontal radiation pattern (HRP)-conjugated secondary antibodies for 2 h at room temperature. Signals were visualized using enhanced chemiluminescence detection reagents (Millipore, Billerica, MA, USA) and imaged using an Image Quant LAS-4000 (Fujifilm, Tokyo, Japan). All the experiments were independent and conducted at least three times. The quantification of proteins was normalized to α-tubulin that was normalized as 1 and then compared to the control group. The value of proteins was determined by software GraphPad prism.

The following antibodies were obtained from Cell Signaling Technology (Beverly, Boston, Massachusett, USA): anti-Bcl-2, anti-Bax, anti-cytochrome c, anti-*α*-tubulin and goat anti-rabbit HRP-conjugated antibodies.

#### 3.3.6. Statistical Analysis

All results were expressed as mean ± SD for three independent experiments. The statistical significance of differences between means was calculated using One-way analysis of variance (ANOVA) by SPSS 19.0. The value of ***p* < 0.01 and **p* < 0.05 were considered statistically significant.

## 4. Conclusions

In summary, novel cytisine-pterocarpan derived compounds were synthesized via DMAP-mediated biomimetic synthetic strategy with a simple, mild, and economic method. Screening of the reaction conditions implied that 1,4-dioxane was the optimal reaction solvent. Under the optimized reaction conditions, **4** was efficiently synthesized with a high yield. Moreover, the cytotoxic activity and possible molecular mechanism of **4** against MDA-MB-231 cell were investigated by MTT, flow cytometry, and Western blot analysis. Overall, the results of MTT assays in the present study suggested that **4** demonstrated high cytotoxic activity against MDA-MB-231 cell and low cytotoxicity against RAW 264.7 and BV2 cell lines. Less damage to RAW 264.7 and BV2 cells indicated that **4** produced less toxicity adverse effects on the central nervous system (Figure 2). The results of flow cytometry assays suggested that the cytotoxic activity of **4** was induced by apoptosis (Figure 3). Moreover, the understanding of apoptosis has provided the basis for novel targeted therapies that induce cell death or sensitize cancer cells to established cytotoxic agents [30].

The mitochondrion-mediated apoptotic signaling pathway serves a critical function in inducing the apoptosis of cancer cells [31]. The Bcl-2, Bax, and cytochrome c proteins are primary regulative proteins in the mitochondrion-mediated apoptotic signaling pathway [32]. The Bcl-2 family of proteins, which shares homology in any of the four common Bcl-2 homology (BH) domains, were highly related with the apoptosis [33]. Bax is able to decrease the inhibitory effect of Bcl-2 on the release of cytochrome c [34]. As shown in Figure 4, the expression of pro-apoptotic proteins, Bax was upregulated significantly in MDA-MB-231 cell lines with the treatment of **4**, On the contrary, the expression of Bcl-2, which are identified as anti-apoptotic proteins, was significantly downregulated with the treatment of **4** in MDA-MB-231 cell. Overall, apoptosis was induced as determined via flow cytometry, and Western blot analysis revealed the downregulation of Bcl-2 and upregulation of Bax, which are important regulators contributing to apoptosis. Therefore, the data suggested that compound **4** inhibited breast cancer progression through inducing apoptosis On the basis of these results, the protein ratio of Bax and Bcl-2 increased compared to the control group (***p* < 0.01). This study indicated that apoptosis might be induced by mitochondrion-mediated apoptotic via regulating the Bax/Bcl-2 ratio and promoting the release of cytochrome c from the mitochondrion to the cytoplasm in MDA-MB-231 cells (Figure 5). 

Previous evidence suggests that pro- and anti-apoptotic activities are tightly regulated by the Bcl-2 family [35]. These proteins regulate cell apoptosis by controlling the release of cytochrome c, through adjusting the permeability of the mitochondrial membrane [36]. Bax is inserted into the mitochondrial membrane and facilitates the release of cytochrome c into the cytoplasm, while Bcl-2 impedes cytochrome c release [37]. Cytochrome c promotes the intrinsic pathway of apoptosis, which is activated after its combination with deoxyadenosine triphosphate (dATP) and apoptotic protease-activating factor (Apaf1) to form an apoptosome [38].

In conclusion, the results of this study indicated that compound **4** is a novel agent for treatment of breast cancer patients. It decreased proliferation and increased apoptosis of MDA-MB-231 cells by reducing activation of Bcl-2 and initiating the mitochondrial apoptosis pathway. It may serve as an effective agent for breast cancer patients.

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
