# Peer review of "Cytisine-Pterocarpan Derived Compounds: Biomimetic Synthesis and Apoptosis-Inducing Activity in Human Breast Cancer Cells"

_molecules, 2018, doi:10.3390/molecules23123059_

Round 1

Reviewer 1 Report

In received to review manuscript (molecules-387040) the authors have developed the method of synthesis of tonkinesisnes A and B and the study of their cytotoxic effects on normal and tumor cell lines. The authors raised in their work an important topic concerning the search for effective anticancer therapies for breast cancer. They also emphasized the importance of apoptosis as the main mechanism for the elimination of cancer cells. The starting compounds on which the authors were based were alkaloids tonkinesisnes A and B with documented biological effects. The authors achieved a biological goal, their compounds were cytotoxic against MDA-MB-231 cell line with IC50 = 19.2 µM. They also proved that cytotoxic activity was induced by apoptosis, which was confirmed by tests with Annexin V and western blot analysis.

The paper deserves to be published in Molecules after corrections. The main claims of the paper are properly placed in the context of the previous literature. The experimental data support the claims. The title of paper reflects the issues of presented work. An abstract represents the achievements of work. Reviewer recommends the paper to reconsideration after major revision that is required.

The main reviewer’s objections are:

1.      First of all, the manuscript should be divided into sections: Introduction, Results and Discussion, Materials and Methods, Conclusion.

2.      First paragraph (page 1, lines 26-35) should be moved to the line 66 on page 2. This change will be beneficial to the introduction soundness.

3.      Table 1 should contain individual yields for 3 and 4, in fact the compound 4 was obtained in yield of 61% in method 4 (entry 4).

4.      Table 2 could contain IC50 data for doxorubicin and CPT-11.

5.      Information from line 118 to 120 should be moved to the figure caption of Figure 3.

6.      Discussion about the results of the western blot analysis must be rewritten. This fragment contains information that is repeated: lines 133-138 were included in the lines 122-130.

7.      Experimental section with method of synthesis, MTT assay, Annexin V-FITC/PI analysis, Western blot analysis should be included in the main text. Spectra and HPLC chromatograms should be attached as Supplement.

8.      The authors did not carry out the correct interpretation of the 1H NMR spectra. For compound 3, proton at 6.4 ppm does not belong to OH group but to H-4. For compound 4, signal at 5.29 ppm does not exist in 1H NMR spectrum (signal  should be removed). The authors should sign protons 14 α, 14 β on the structure of tonkinesine A as well as tonkinesine B. Signal at 3.51 ppm (compd 3) belongs to H-14 α, not to H-10 a.

9.      Rewiever suggests changing the title to Cytisine-Pterocarpan Derived Compounds - Biomimetic Synthesis and apoptosis-inducing activity in Human Breast Cancer Cells

Author Response

Dear Reviewer,

Thank you for providing us useful comments for our manuscript, which have helped us greatly improve the quality of the manuscript The point-by-point replies are as follows.

We are looking forward to hearing from you.

1. First of all, the manuscript should be divided into sections: Introduction, Results and Discussion, Materials and Methods, Conclusion.

We thank the reviewer for the suggestion. The manuscript has been revised according to the suggestion.

2. First paragraph (page 1, lines 26-35) should be moved to the line 66 on page 2. This change will be beneficial to the introduction soundness.

We thank the reviewer for the suggestion. It has been moved (page 1, lines 26-35) to the line 66 on page 2 in first paragraph as suggested.

3. Table 1 should contain individual yields for 3 and 4, in fact the compound 4 was obtained in yield of 61% in method 4 (entry 4).

We thank the reviewer for the suggestion. The individual yields for 3 and 4 have been added in Table 1 as suggested.

4. Table 2 could contain IC50 data for doxorubicin and CPT-11.

We thank the reviewer for the suggestion. IC50 data for doxorubicin treating with MCF-7 and MDA-MB-231 cell have been added in Table 2 as suggested.

5. Information from line 118 to 120 should be moved to the figure caption of Figure3.

We thank the reviewer for the suggestion. The manuscript has been revised according to the suggestion.

6. Discussion about the results of the western blot analysis must be rewritten. This fragment contains information that is repeated: lines 133-138 were included in the lines 122-130.

We thank the reviewer for the suggestion. The discussion about the results of the western blot analysis have been added in the manuscript as suggested.

7. Experimental section with method of synthesis, MTT assay, Annexin V-FITC/PI analysis, Western blot analysis should be included in the main text. Spectra and HPLC chromatograms should be attached as Supplement.

We thank the reviewer for the suggestion. The manuscript has been revised according to the suggestion.

8. The authors did not carry out the correct interpretation of the 1H NMR spectra. For compound 3, proton at 6.4 ppm does not belong to OH group but to H-4. For compound 4, signal at 5.29 ppm does not exist in 1H NMR spectrum (signal should be removed). The authors should sign protons 14α, 14β on the structure of tonkinesine A as well as tonkinesine B. Signal at 3.51 ppm (compd 3) belongs to H-14α, not to H-10 a.

We thank the reviewer for the suggestion. The manuscript has been revised according to the suggestion.

9.Rewiever suggests changing the title to Cytisine-Pterocarpan Derived Compounds - Biomimetic Synthesis and apoptosisinducing activity in Human Breast Cancer Cells

We thank the reviewer for the suggestion. The title has been changed as suggested.

Sincerely

Xing-Nuo Li, PhD

College of Pharmaceutical Science, Zhejiang University of Technology,

China.

Hangzhou, 310014

Email: li_xingnuo@163.com

Phone: +86-571-88320613

Reviewer 2 Report

The authors have synthesised 2 compounds (3 and 4) and 4 has been tested together with starting materials (1 and 2) as anticancer agents. The activity of compound 4 showed good activity with low tocicity. Before article can be suitable for publishing in Molecules several points need to be properly addressed. The title of the article is too general (Cytisine-Pterocarpan Derived Compounds) becasue only one compound is synthesised and tested (compound 3 also isolated but not tested). In experimental data (synthesis of 4) there is no data about solvent used for TLC monitoring.

Author Response

Dear Reviewer,

Thank you for providing us with useful comments on our manuscript. Below is our

response to your comments. We look forward to hearing from you soon.

1. The title of the article is too general (Cytisine-Pterocarpan Derived Compounds)

becasue only one compound is synthesised and tested (compound 3 also isolated but

not tested). In experimental data (synthesis of 4) there is no data about solvent used

for TLC monitoring.

We thank the reviewer for the suggestion. The title of article has been changed

according to the reviewer 1’s comment (Cytisine-Pterocarpan Derived Compounds -

Biomimetic Synthesis and apoptosis-inducing activity in Human Breast Cancer Cells).

The data of solvent used for TLC monitoring have been added in the manuscript.

Sincerely

Xing-Nuo Li, PhD

College of Pharmaceutical Science, Zhejiang University of Technology,

China.

Hangzhou, 310014

Email: li_xingnuo@163.com

Phone: +86-571-88320613

Reviewer 3 Report

General comments:

The authors claimed that compound 4 might induce mitochondrion-mediated apoptosis via regulating the ratio of Bax/Bcl-2 and promoting the release of cytochrome c from the mitochondrion to the cytoplasm in MDA-MB-231 cell.

Major comments:

1. Although several evidences shows that compound 4 might induce mitochondrion-mediated apoptosis, they lacked of the apoptosis inhibitor control. In other word, at least annexin V/PI (Fig. 3) or western blotting for Bcl2 or Bax (Fig. 4) should be added with or without apoptosis inhibitor such as Z-VAD-Fmk to validate the induction expression is apoptosis.

Minor comments:

1. annexin V/PI should provide the full name as such as propidium iodide.

Author Response

Dear Reviewer,

Thank you for providing us with useful comments on our manuscript. Below is our

response to your comments. We are looking forward to hearing from you.

Major comments:

1. Although several evidences shows that compound 4 might induce

mitochondrion-mediated apoptosis, they lacked of the apoptosis inhibitor control. In

other word, at least annexin V/PI (Fig.3) or western blotting for Bcl2 or Bax (Fig. 4)

should be added with or without apoptosis inhibitor such as Z-VAD-Fmk to validate

the induction expression is apoptosis.

We thank the reviewer for the suggestion. As we can see from the data of flow

cytometry, the doses we choose (6.25- and 12.5-μM) can cause the partially early- and

late- apoptosis. And furthermore, western blotting result identified this apoptosis was

attribute to mitochondrion pathway. We agree that lacking of the apoptosis inhibitor

control could not validate the induction expression is apoptosis, but the results can

partly reflect the molecular mechanisms and this method was also used to investigate

the signaling pathway of apoptosis in some high-level articles, such as that shown

below.

1. Viacava F.; Jerry E.C.; John C.F.; Mi-Kyung Y.; Christy R.G.; Amanda N.;

Katherine B.; Li O.; Lie M.; Stephen W.W.; Douglas R.G.; Richard W.K.; PUMA

binding induces partial unfolding within BCL-xL to disrupt p53 binding and promote

apoptosis. Nature Chemical Biology. 2013, 9, 163-172.

2. Evripidis G.; Denis e R.; Joseph A.B.; Elizaveta S.L.; Loren D.W.; Direct and

selective small-molecule activation of proapoptotic Bax. Nature Chemical Biology.

2012, 8, 1-7.

Minor comments:

1. annexin V/PI should provide the full name as such as propidium iodide.

The full name of annexin V/Propidium Iodide had been included in the manuscript.

Sincerely

Xing-Nuo Li, PhD

College of Pharmaceutical Science, Zhejiang University of Technology,

China.

Hangzhou, 310014

Email: li_xingnuo@163.com

Phone: +86-571-88320613

Reviewer 4 Report

In this manuscript, Penga et al. have investigated the effect of a synthetic compound on human breast cancer cell lines to be considered a novel therapeutic approach. They have demonstrated that this compound plays an inhibitory role by reducing viability of these cells which is mediated through regulation of intrinsic arm of apoptosis in the mitochondria. Although the study is well designed and executed addressing an unmet therapeutic need, I have the following concerns to be addressed before acceptance for publication.

1.      The importance of intrinsic mitochondrial-mediated apoptosis in development of breast tumor and targeting it as novel therapies against breast cancer should be explained in Introduction.

2.      The rationale behind choosing different breast cancer cell lines needs to be explained in terms of the cell origin, pathological features, etc.

3.      Statistical significance of the data must be shown with stars in each graph (Fig. 2).

4.      The authors should explain why 6.25- and 12.5-mM concentrations were used in apoptosis experiments while IC50 and optimal doses in viability assay (Fig. 2) are different.

5.      Similar concern for choosing the time point (48h). I would recommend testing two more time points to understand the kinetic response (24 and 72h). Then, why signaling Western blot studies have been carried out on the protein lysates harvested 12h after treatment?

6.      In Fig. 4, Bax blot is faint and the difference cannot be appreciated at all. Please improve the quality of this blot. Otherwise, this data is not convincing for publication as it is.

7.      The Discussion should be rewritten by further explanation of the findings, their significant to the field, critical interpretation according to strong literature review, and providing detailed future directions.

8.      In Fig. 3D, 4A2, 4B2, 5C2, and 5C3, it is not clear which groups are compared to each other. Why there are stars representing P values on control group? Please draw a line on the groups which are being compared to each other.

9.      Major grammatical errors to be edited by a professional native English.

Author Response

Dear Reviewer,

Thank you for providing us with useful comments on our manuscript. Below is our

response to your comments. We are looking forward to hearing from you.

1. The importance of intrinsic mitochondrial-mediated apoptosis in development of

breast tumor and targeting it as novel therapies against breast cancer should be

explained in Introduction.

We thank the reviewer for the suggestion. We had included the importance of intrinsic

mitochondrial-mediated apoptosis in development of breast tumor and targeting it as

novel therapies against breast cancer in Introduction.

2. The rationale behind choosing different breast cancer cell lines needs to be

explained in terms of the cell origin, pathological features, etc.

We had included the cell origin and cell culture as suggested.

MCF-7 and MDA-MB-231 are human breast cancer cell lines with low and high

metastatic potential, respectively. MDA-MB-231 is highly metastatic malignant breast

cancer cell line and is easy to form tumors. It is often used for tumor metastasis

research, especially lung metastasis. Meanwhile, MCF-7 is an orthotopic ER-positive

breast cancer cell line with a lower degree of malignancy than MDA-MB-231. It is

not so easy to form tumors, but it can be used as a good animal tumor model under

estrogen stimulation.

3. Statistical significance of the data must be shown with stars in each graph (Fig. 2).

The statistical significance of the data in Fig.2 has been added with stars.

4. The authors should explain why 6.25- and 12.5-μM concentrations were used in

apoptosis experiments while IC 50 and optimal doses in viability assay (Fig. 2) are

different.

It is usually recognized that the concentration which is below IC 50 is proper to

distinguish the early apoptosis and late apoptosis. As we can see from the data of flow

cytometry, the doses we choose (6.25- and 12.5-μM) can already cause the partially

early- and late- apoptosis. In that case, if we use higher doses, necrosis may happen

and all the results may be meaningless.

5. Similar concern for choosing the time point (48h). I would recommend testing two

more time points to understand the kinetic reponse (24 and 72h). Then, why signaling

Western blot studies have been carried out on the protein lysates harvested 12h after

treatment?

Different time points were used in the protein lysates collection, and we found the

experiment exhibited better results of expression of proteins when treated for 12h.

That may attribute to higher doses may lead to necrosis of the cells and can hardly be

tested with WB apoptosis parameters.

6. In Fig. 4, Bax blot is faint and the difference cannot be appreciated at all. Please

improve the quality of this blot. Otherwise, this data is not convincing for publication

as it is.

We thank the reviewer for the suggestion. Though the expression of Bax is not very

significant, the quantification of the ratio of Bax/Bcl-2 is quite significant. And the

quantification for the ratio of Bax/Bcl-2 in Figure 4 had been made into a bar chart

named figure 4C.

7. The Discussion should be rewritten by further explanation of the findings, their

significant to the field, critical interpretation according to strong literature review,

and providing detailed future directions.

We thank the reviewer for the suggestion. The further explanation of our findings,

critical interpretation and detailed future directions had been rewritten in discussion.

8. In Fig. 3D, 4A2, 4B2, 5C2, and 5C3, it is not clear which groups are compared to

each other. Why there are stars representing P values on control group? Please draw

a line on the groups which are being compared to each other.

We had corrected the mistake of Fig. 3D, 4A2, 4B2, 4C, 5D2, and 5D3, and we had

indicated the control group in Fig. 3D, 4A2, 4B2, 4C, 5D2, and 5D3.

9. Major grammatical errors to be edited by a professional native English.

We thank the reviewer for the suggestion. These errors have been corrected.

Sincerely

Xing-Nuo Li, PhD

College of Pharmaceutical Science, Zhejiang University of Technology,

China.

Hangzhou, 310014

Email: li_xingnuo@163.com

Phone: +86-571-88320613

Round 2

Reviewer 1 Report

The authors complied with all comments. I accept the manuscript in present form.

A small suggestion: Line 172 - The title of the subsection 3.2.does not need "(Scheme 1)".

Reviewer 3 Report

All reviewer's concerns had been well responded.

Reviewer 4 Report

My comments have been properly addressed except Bax Western blotting. I would recommend removing this data and associated text. 

There are still grammatical errors which need to be corrected before publication, e.g. Pages 144-145: "were"  instead of "was". 

Good luck